# Design and Optimization of Coil for Transcutaneous Energy Transmission System

**Ruiming Wu** [1,2,*] , **Haonan Li** [1,2] , **Jiangyu Chen** [1,2] , **Qi Le** [1,2] , **Lijun Wang** [3] , **Feng Huang** [1,2] and **Yang Fu** [1,2,*]

[1] School of Mechanical and Energy Engineering, Zhejiang University of Science and Technology, Hangzhou 310023, China; rjlhn@zust.edu.cn (H.L.); jiangyuchen@zust.edu.cn (J.C.); qile@zust.edu.cn (Q.L.); hf@zust.edu.cn (F.H.)
[2] Zhejiang Provincial Key Laboratory of Food Logistics Equipment and Technology, Hangzhou 310023, China
[3] School of Automation and Electrical Engineering, Zhejiang University of Science and Technology, Hangzhou 310023, China; dagouwang@zju.edu.cn
* Correspondence: wuruiming@zust.edu.cn (R.W.); yangfu@zust.edu.cn (Y.F.)

**Abstract:** This article presents a coil couple-based transcutaneous energy transmission system (TETS) for wirelessly powering implanted artificial hearts. In the TETS, the performance of the system is commonly affected by the change in the position of the coupling coils, which are placed inside and outside the skin. However, to some extent, the influence of coupling efficiency caused by misalignment can be reduced by optimizing the coil. Thus, different types of coils are designed in this paper for comparison. It has been found that the curved coil better fits the surface of the skin and provides better performance for the TETS. Various types of curved coils have been designed in response to observed bending deformations, dislocations, and other coupling variations in the curved coil couple. The numerical model of the TETS is established to analyze the effects of the different types of coils. Subsequently, a series of experiments are designed to evaluate the resilience to misalignment and to verify the heating of the coil under conditions of severe coupling misalignment. The results indicated that, in the case of misalignment of the coils used in artificial hearts, the curved transmission coil demonstrated superior efficiency and lower temperature rise compared to the planar coil.

**Keywords:** transcutaneous energy transmission system; artificial hearts; curved coil; coil optimization

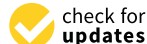



## 1. Introduction

The artificial heart, as an advanced medical device, is utilized for treating patients with heart failure. It effectively replaces or supports the failing biological heart to restore its function, significantly avoiding issues related to donor shortages and post-transplant immune rejection commonly associated with heart transplantation strategies [1–3]. Moreover, a continuous and stable energy supply is one of the critical factors limiting the development of artificial hearts. In particular, the question of how to safely and efficiently power the implanted artificial heart remains crucial. Wireless power transfer technology is regarded as an effective solution to ensure stable power supply to artificial hearts while eliminating the need for transcutaneous wires, thereby significantly reducing the risk of infection. The adoption of wireless power transfer technology enhances the freedom and comfort of patients using artificial heart devices, offering them a higher quality of life during treatment [4–6].

In practical applications, coil misalignment presents a significant technical challenge that affects the efficiency and safety of wireless energy transfer systems [7,8]. This misalignment typically occurs between the external coil (transmitter) and the internal coil (receiver) implanted within the body. Everyday activities of patients, such as walking, turning, or bending, may change the relative positions of the externally mounted coil and

the internal coil, leading to physical misalignment—a common cause of coil misalignment. Wireless energy transfer systems rely on close magnetic coupling between coils to efficiently transmit energy. When the relative positions of the coils change, the effectiveness of the magnetic coupling is significantly reduced, resulting in decreased energy transmission efficiency [9,10]. Additionally, coil misalignment may cause the magnetic field to concentrate in areas where it should not, increasing the temperature of local tissues, potentially leading to thermal injury or other biological effects, and thus increasing the risk associated with use.

In previous research efforts aimed at enhancing the transmission characteristics of wireless power transfer systems, researchers have primarily focused on optimizing coil structures and circuit designs. Regarding coil optimization, the use of a multi-coil structure has proven effective in stabilizing system transmission and achieving a more uniform magnetic field distribution. However, the coupling level varies between different positioned transmitter and receiver coils, and activating all coils simultaneously can lead to unnecessary energy losses and the issue of cross-coupling among multiple coils should not be overlooked [11–13]. Additionally, by altering the shape and structure of the coils, such as using DD-type coils, DDQ-type coils, multi-layer stacked coils, and 3D orthogonal coils, a uniform and symmetric magnetic field is formed, reducing the system's sensitivity to coil misalignment [14–16]. Considering the materials for coils, some scholars have suggested using superconducting materials due to their low-loss characteristics, which can reduce ohmic losses in energy transmission and improve the quality factor of the resonant coils, thereby enhancing the system's transmission efficiency. However, superconducting materials have inherent limitations, including strict temperature requirements that prevent widespread adoption, and are still in the exploratory stage [17–20]. In terms of circuit optimization design, complex compensation networks have been employed to optimize the topology of the circuit, enabling stable output power despite changes in the coupling coefficient [21–23]. Furthermore, by adding impedance-matching circuits and feedback adjustment systems, the dynamic power adjustment capability of the system is effectively enhanced, providing good resistance to misalignment. However, the additional sensors and circuits undoubtedly increase the complexity of the system, which is not conducive to the implantation of miniaturized medical devices [24,25].

This study, taking into account the implantation environment of artificial hearts and the potential for coil misalignment caused by patients' daily activities, analyzes and optimizes the anti-malposition capabilities of the transcutaneous energy transfer system (TETS) from the perspectives of coupling coefficient and transmission efficiency by optimizing key parameters of the coil. Moreover, to better conform to the three-dimensional curved structure of the human body, this paper proposes the use of curved coils that offer spatial adaptability and robust anti-malposition characteristics. Through temperature simulation and experimental validation, it has been demonstrated that curved coils operate with lower temperature increases, thereby significantly enhancing the system's biological safety. Compared to traditional planar coils, curved coils, due to their better conformity to the human body, improve patient comfort and acceptance. These advantages make curved coils a crucial direction for the design of future wireless charging systems.

## 2. System Theoretical Analysis

The fundamental structure of the transcutaneous energy transmission system based on a coil coupler proposed in this paper is illustrated in Figure 1. It comprises a DC supply, an inverter circuit, resonance compensation circuits, a coil coupler, a rectification and filtering circuit, and a load. The system is powered by an external DC source, which supplies the necessary alternating current via a full-bridge inverter circuit. The primary transmitting coil converts electrical energy into magnetic energy, which is radiated outward. The secondary coil transforms the received electromagnetic energy back into electrical energy, which is then supplied to the artificial heart after passing through the rectification and filtering circuit. The equivalent dual-coil circuit topology model of the system is shown in Figure 2.

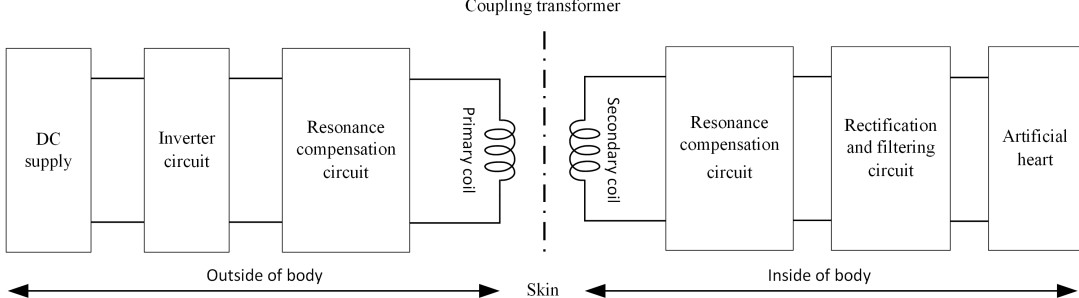

**Figure 1.** TETS structural diagram.

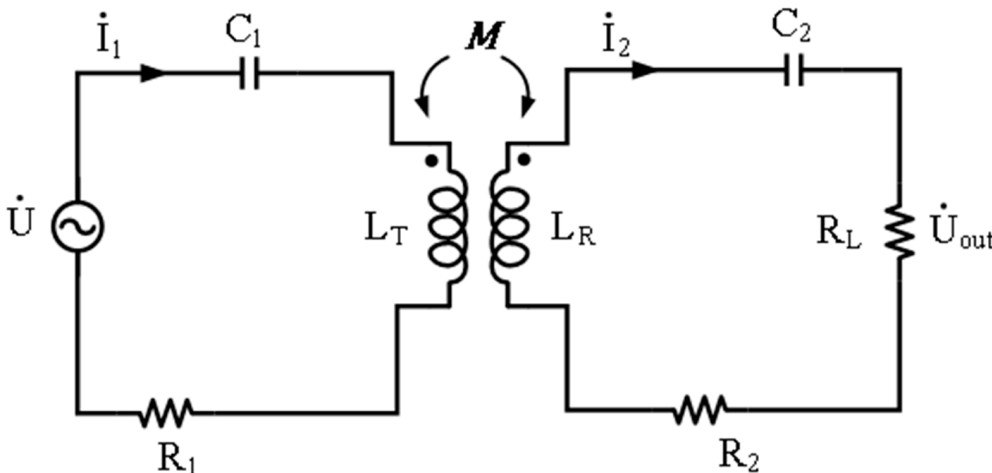

**Figure 2.** Equivalent circuit of a two-coil wireless power transmission.

As depicted in Figure 2, $U$ denotes the input voltage at the external transmission side, while $U_{out}$ represents the output voltage at the internal reception side. The capacitors $C_1$ and $C_2$ serve as resonance compensation capacitors on the external transmission and internal reception sides, respectively. $R_1$ is the equivalent internal resistance of the transmitting coil, and $R_2$ is that of the receiving coil. $L_T$ and $L_R$ are the self-inductances of the transmitting and receiving coils, respectively. $R_L$ is the resistance of the load, and $M$ denotes the mutual inductance between the coils.

Based on Kirchhoff's Voltage Law, the mesh equations for the circuit can be derived:

$$\begin{cases} \left(R_1 + \frac{1}{j\omega C_1} + j\omega L_T\right)\dot{I}_1 - j\omega M \dot{I}_2 = \dot{U} \\ \left(R_L + R_2 + \frac{1}{j\omega C_2} + j\omega L_R\right)\dot{I}_2 - j\omega M \dot{I}_1 = 0 \end{cases} \tag{1}$$

where $f$ represents the frequency in Hertz (Hz); $\omega$ denotes the angular frequency in radians per second (rad/s).

According to the vector method, the input impedances of the transmitting side circuit, $Z_1$, and the receiving side circuit, $Z_2$, are given as follows:

$$\begin{cases} R_1 + \frac{1}{j\omega C_1} + j\omega L_T = Z_1 \\ R_L + R_2 + \frac{1}{j\omega C_2} + j\omega L_R = Z_2 \end{cases} \tag{2}$$

The total input impedance of the system, $Z_{\text{in}}$, is calculated to be

$$Z_{\text{in}} = \frac{\dot{U}}{\dot{I}_1} = R_1 + j\omega L_T + \frac{1}{j\omega C_1} + \frac{\omega^2 M^2}{R_L + R_2 + \frac{1}{j\omega C_2} + j\omega L_R} = Z_1 + \frac{\omega^2 M^2}{Z_2} \tag{3}$$

When the transmission system is in resonance, the imaginary part of Equation (2) becomes zero, that is

$$\begin{cases} \frac{1}{j\omega C_1} + j\omega L_T = 0 \\ \frac{1}{j\omega C_2} + j\omega L_R = 0 \end{cases} \tag{4}$$

Thus, Equation (3) can be simplified to

$$Z_{\text{in}} = R_1 + \frac{\omega^2 M^2}{R_L + R_2} \tag{5}$$

The currents $\dot{I}_1$ and $\dot{I}_2$ in the circuits on both sides can be calculated as follows:

$$\begin{cases} \dot{I}_1 = \frac{\dot{U}}{Z_{\text{in}}} = \frac{\dot{U}(R_L+R_2)}{R_1(R_L+R_2)+\omega^2 M^2} \\ \dot{I}_2 = \frac{j\omega M \dot{I}_1}{R_L+R_2} = \frac{j\omega M \dot{U}}{R_1(R_L+R_2)+\omega^2 M^2} \end{cases} \tag{6}$$

Furthermore, the expressions for the system's output power $P$ and efficiency $\eta$ can be derived as follows:

$$\begin{cases} P = I_2^2 \cdot R_L = \frac{\omega^2 M^2 U^2 R_L}{[R_1(R_L+R_2)+\omega^2 M^2]^2} \\ \eta = \frac{P}{U \cdot I_1} = \frac{\omega^2 M^2 R_L}{R_1(R_L+R_2)^2+\omega^2 M^2(R_L+R_2)} \end{cases} \tag{7}$$

From Equation (7), it is evident that the influences on the system's power and efficiency are multifaceted. When the system's frequency, load, and input voltage are fixed, the performance parameters of the two resonant coils largely determine the power and efficiency of the system. The mutual inductance $M$ of the coils is closely associated with the coupling coefficient $k$ between the two coils, adhering to the following equation:

$$k = \frac{M}{\sqrt{L_T L_R}} \tag{8}$$

The coupling coefficient $k$ can be used to indicate the degree of coupling between two coils; a higher $k$ implies greater energy transmission efficiency and less energy loss. Therefore, optimizing the coupling coefficient is crucial in wireless energy transmission systems. When there is a misalignment between the coils, both the coupling coefficient and mutual inductance decrease to varying extents, which in turn reduces the transmission efficiency. Consequently, from the perspective of coil optimization, enhancing the coupling coefficient can improve the system's resilience to misalignment and overall transmission efficiency.

### 3. Optimization of Coils

Considering the spatial constraints of the artificial heart implantation environment, the receiving coil is embedded within the human thoracic cavity, and the transmitting coil is placed close to the surface of the thoracic cavity. Therefore, the wireless energy transmission system generally employs planar spiral coils rather than volumetric spiral coils. This study aims to enhance the coupling coefficient between coils and the system's transmission efficiency, optimizing the parameters of the transmission coils using the COMSOL finite element simulation software. Due to the numerous variable parameters of both coils, it is challenging to optimize the parameters of both the transmitting and receiving coils simultaneously. Hence, this research focuses on the parameter optimization of the transmitting coil located outside the body. The structure of the planar coil is shown in Figure 3, where $D_{out}$ represents the coil's outer diameter, $D_{in}$ is the coil's inner diameter, $D_{wire}$ is the diameter of the Litz wire, and $D_{pitch}$ is the pitch of turns. In the simulations and subsequent experiments of this section, the specifications of the receiving coils are all Litz wires, composed of 400 strands, each with a diameter of 0.04 mm. The receiving coil has a wire diameter of 1.1 mm, an outer diameter of 60 mm, an inner diameter of 20 mm, 15 turns, and a pitch of turns of zero.

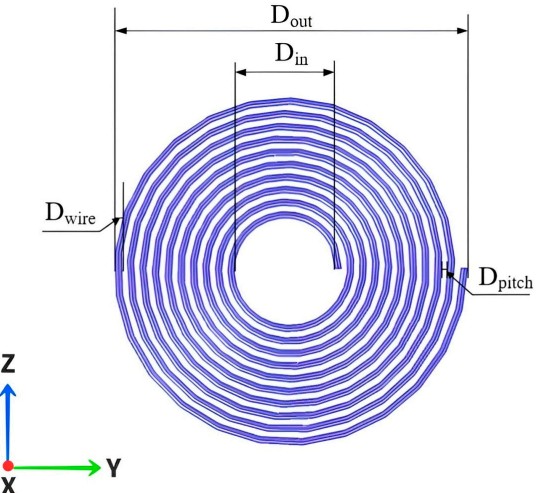

**Figure 3.** Planar spiral coil structure diagram.

According to Reference [26], the inductance of the coil can be obtained using the surface current method:

$$L = \frac{\mu_0 N^2 D_{avg} c_1}{2} \left( ln\frac{c_2}{\rho} + c_3\rho + c_4\rho^2 \right) \tag{9}$$

In Equation (9), $c_1$, $c_2$, $c_3$, and $c_4$ are fitting parameters for a circular structure with values $c_1 = 1.00$, $c_2 = 2.46$, $c_3 = 0.00$, and $c_4 = 0.20$; $\mu_0$ is the permeability of free space, valued at $4\pi \times 10^{-7}$ $T \cdot m/A$; $N$ represents the number of turns of the coil; $D_{avg}$ is the average diameter of the coil, calculated as $D_{avg} = 0.5 \times (D_{out} + D_{in})$; and $\rho$ is the fill factor, defined as $\rho = (D_{out} - D_{in})/(D_{out} + D_{in})$.

According to Reference [27], the quality factor $Q$ of a coil constitutes a crucial parameter, intimately linked to the magnitude of losses within the coil. $Q = \omega L/R$, and the maximum coil efficiency $\eta_{max}$ can be expressed as

$$\eta_{max} = \frac{k^2 Q_1 Q_2}{\left(1 + \sqrt{1 + k^2 Q_1 Q_2}\right)^2} \tag{10}$$

In Equation (10), $k$ is the coupling coefficient; $Q_1$ and $Q_2$ are the quality factors of the transmitting coil and the receiving coil.

*3.1. Optimal Coil Inner Diameter*

Considering the spatial constraints of the human thoracic cavity, the maximum outer diameter $D_{out}$ of the transmission coil is generally restricted to 60 mm, while the inner diameter $D_{in}$ can vary. To explore the impact of the transmitting coil's inner diameter on the Transcutaneous Energy Transmission Systems (TETSs), the simulation was set with an *X*-axis misalignment distance of 15 mm and analyzed from the perspective of mutual inductance and the coupling coefficient between the coils is conducted, as shown in Figure 4.

As depicted in Figure 4, it can be observed that the mutual inductance and coupling coefficient between the coils exhibit opposite trends with the variation in the emitter coil's inner radius. Initially, the mutual inductance slightly increases with the change in inner radius and then decreases, whereas the coupling coefficient rapidly increases at first and then stabilizes. At an inner radius of 15 mm, the curves of mutual inductance and coupling coefficient intersect, at which point both values reach relatively high levels. Specifically, as the inner radius increases from 5 mm to 15 mm, the mutual inductance gradually rises to a peak, then significantly decreases as the inner radius further increases to 25 mm. Conversely, the coupling coefficient increases rapidly initially, reaching a peak, and then gradually

stabilizes. Analyzing the data and trend lines in Figure 4 reveals that an inner radius of 15 mm maintains both a high mutual inductance and an optimal coupling coefficient. Therefore, considering the mutual inductance and coupling coefficient at different inner radii, 30 mm was ultimately selected as the optimized inner diameter for the coil.

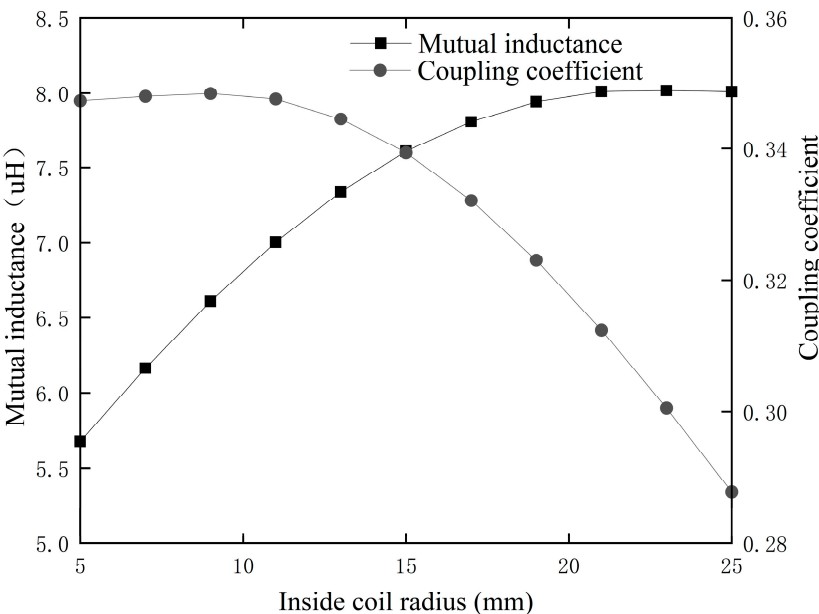

**Figure 4.** Relationship between mutual inductance and coupling coefficient of coils.

### 3.2. Optimal Coil Turn

As one of the main parameters of a coil, an increase in the number of turns implies a larger coil area, which in turn expands the range of magnetic field radiation. To investigate the impact of coil turn number on the system, this paper uses finite element simulation software to calculate an optimal number of turns. In the simulations, the transmission distance of 15 mm in the *X*-axis direction was set and the number of turns in the transmitting coil was varied from 10 to 30 in increments of 5 turns, with an inner diameter of 30 mm for the coil. The coupling coefficients between the two coils are shown in Figure 5.

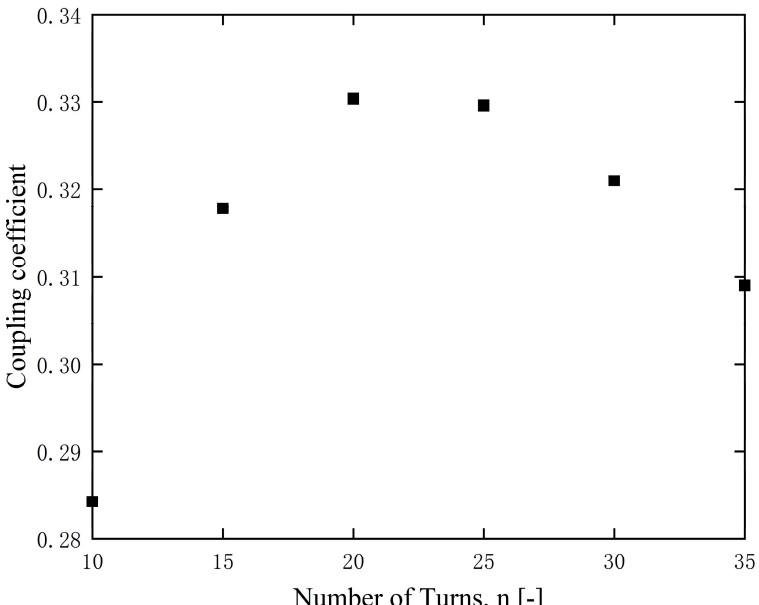

**Figure 5.** Relationship between coupling coefficient and number of turns.

As observed from Figure 5, the coupling coefficient initially increases and then decreases as the number of turns increases, with notably higher values at 20 and 25 turns. This pattern indicates that having more turns is not always beneficial for the system. To better compare the changes in the coupling coefficients at 20 and 25 turns, the simulation was set with an *X*-axis misalignment distance of 15 mm and the *Y*-axis misalignment distance varied between the two coils. The results, as shown in Figure 6, indicate that the coil with 25 turns exhibits a significantly smaller variation in coupling coefficient under misalignment conditions compared to the coil with 20 turns. Consequently, 25 turns was selected as the optimal number of turns for the coil.

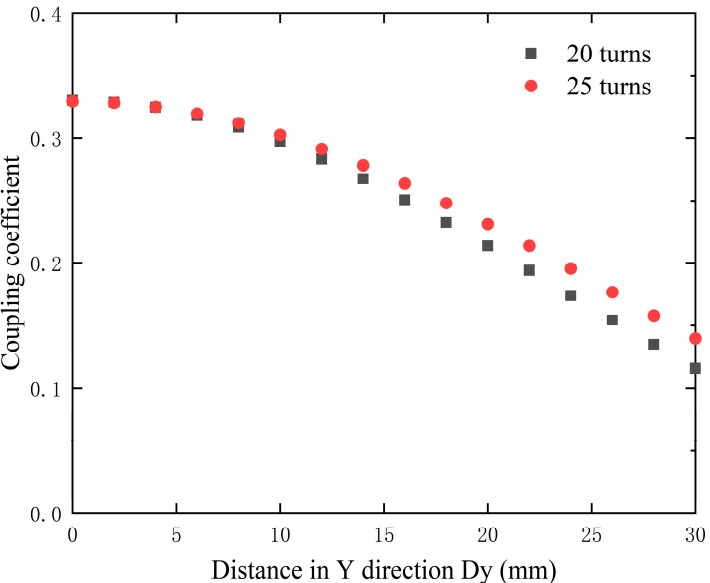

**Figure 6.** Coupling coefficients under *Y*-axis misalignment.

### 3.3. Optimal Pitch of Turns

To explore the relationship between the pitch of turns and the coupling coefficient, initial coil modeling was conducted using finite element simulation software. The inner diameter of the coil was set at 30 mm with 25 turns. The *X*-axis misalignment distance was 15 mm, and the pitches of the turns were set at 0 mm, 1 mm, 2 mm, 3 mm, and 4 mm. Figure 7 illustrates the changes in the coupling coefficient when *Y*-axis misalignment occurs.

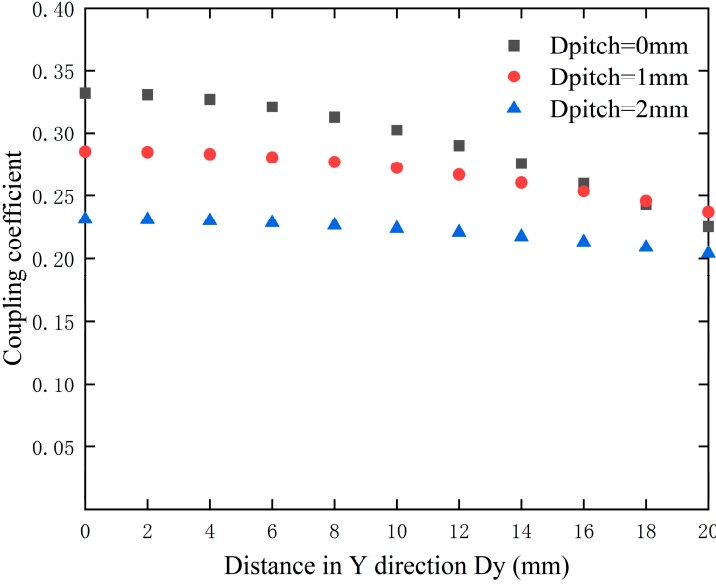

**Figure 7.** Relationship between coupling coefficient and $D_{pitch}$ of turns in coils.

As shown in Figure 7, the coupling coefficient decreases as the misalignment distance increases. The decrease in coupling coefficient is most pronounced when the pitch of turns is 0 mm; however, within a misalignment distance range of 20 mm, the coupling coefficient is superior to that of coils with 1 mm and 2 mm pitch. Therefore, a pitch of 0 mm was selected as the optimal pitch of turns for the coil.

### 3.4. Optimal Bend Degree

In this study, the optimized coils are applied to the TETS, and to better conform to the human thoracic cavity, it is proposed to use curved coils instead of planar coils. Initially, it is necessary to define the degree of curvature of the transmitting coil, as shown in Figure 8, where the angle $\alpha$ between the tangents at the ends of the coil is used to define the coil bending degree.

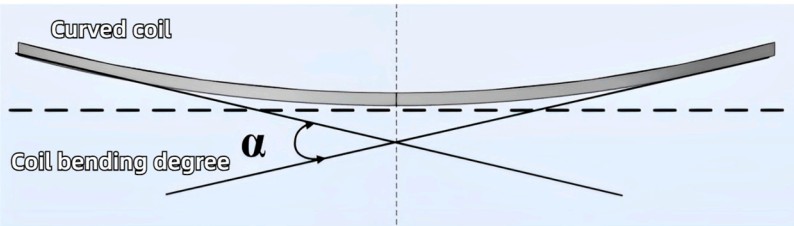

**Figure 8.** Schematic diagram of coil bending degree.

To better demonstrate the changes in the coupling coefficient of the curved coil when *Y*-axis misalignment occurs, in the simulation, the *X*-axis misalignment distance was set to 15 mm with a *Y*-axis misalignment range of 0–30 mm. The changes in the coupling coefficient for coils at various angles of curvature are shown in Figure 9.

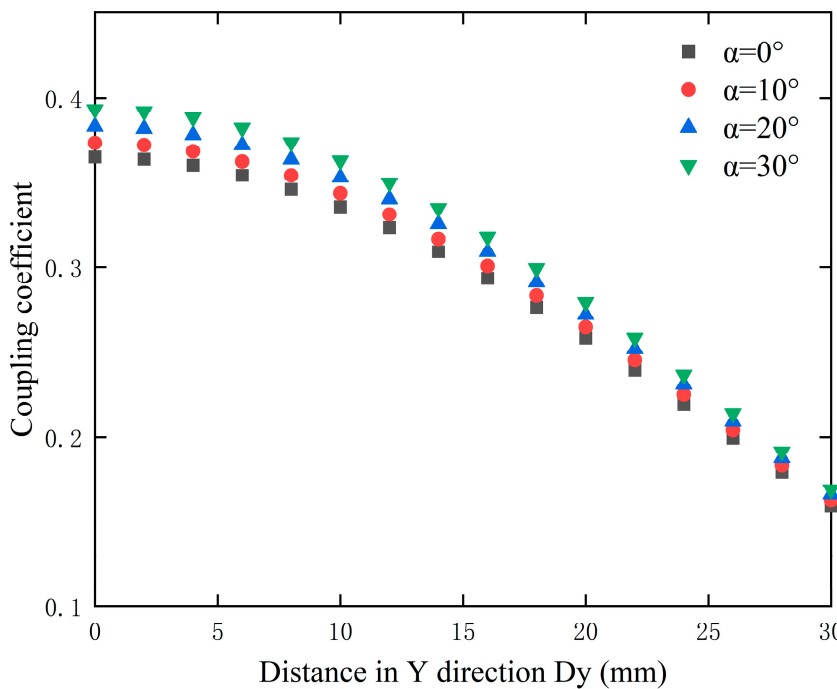

**Figure 9.** Relationship between coil bending degree and coupling coefficient.

As shown in Figure 9, it can be observed that as the misalignment distance along the *Y*-axis increases, and the coupling coefficient for all four degrees of coil curvature exhibits a decreasing trend, regardless of the curvature degree. This indicates that the magnetic coupling strength between the coils weakens with increasing misalignment distance. Fur-

thermore, an analysis of the coupling coefficients for different coil curvatures shows that the 30-degree curved coil consistently maintains the highest coupling coefficient across the entire range of misalignment distances. This suggests that the 30-degree curved coil can provide stronger magnetic coupling at various misaligned distances, thus demonstrating the best anti-malposition performance. Conversely, the 0-degree planar coil maintains the lowest coupling coefficient throughout the misaligned range, indicating that planar coils exhibit weaker coupling performance and are more sensitive to misalignment. Additionally, the 10-degree and 20-degree curved coils, which lie between the 0-degree planar coil and the 30-degree curved coil, exhibit a relatively moderate decline in coupling coefficient with increasing misaligned distance but still do not match the performance of the 30-degree curved coil. The coupling condition of the transmission coils becomes weaker with increasing misalignment. As the misalignment increases, the enhancement effect of the curved coil diminishes accordingly. Therefore, comprehensive analysis indicates that curved coils outperform planar coils, offering better anti-malposition performance.

### 3.5. Temperature Simulation of Coupling Coils

In a wireless power transmission system, the process of transferring energy from the transmitting coil to the receiving coil is accompanied by some energy loss. Part of this energy is dissipated as thermal energy, thereby leading to an increase in coil temperature. Human tissues are highly sensitive to temperature; excessive heat can cause tissue damage or even necrosis. Therefore, when applying wireless energy transmission technology to artificial hearts, it is essential to verify the heating conditions of the transmission coils. In the COMSOL simulation software, two sets of different transmitting coils were simulated: a 30-degree curved coil and a 0-degree planar coil. Both sets of transmitting coils had an inner diameter of 30 mm and 25 turns. The receiving coils had an inner diameter of 30 mm and consist of planar coils with 15 turns. The spacing between all coil turns was 0 mm, and the coils were excited by a circuit current.

Additionally, a bioheat transfer module was added to simulate the heating of the 30-degree curved coil and the 0-degree planar coil under actual working conditions, as well as the temperature rise of surrounding tissues. Based on the actual distribution of human tissues, the simulations in COMSOL were configured with a skin layer of 5 mm, a fat layer of 10 mm, and a muscle layer of 40 mm for both sets. Due to the symmetry of human tissues and coils, a two-dimensional symmetric simulation in COMSOL was employed. Furthermore, identical parameters were set for both simulations, the initial temperature of human tissues and blood was set at 36.5 °C, ignoring the impact of human metabolism, with the metabolic heat source set at 0 W/m$^3$, blood density at 1000 kg/m$^3$, and the external air temperature at 30 °C. Figure 10 shows the temperature rise of the coils after 30 min of electrification.

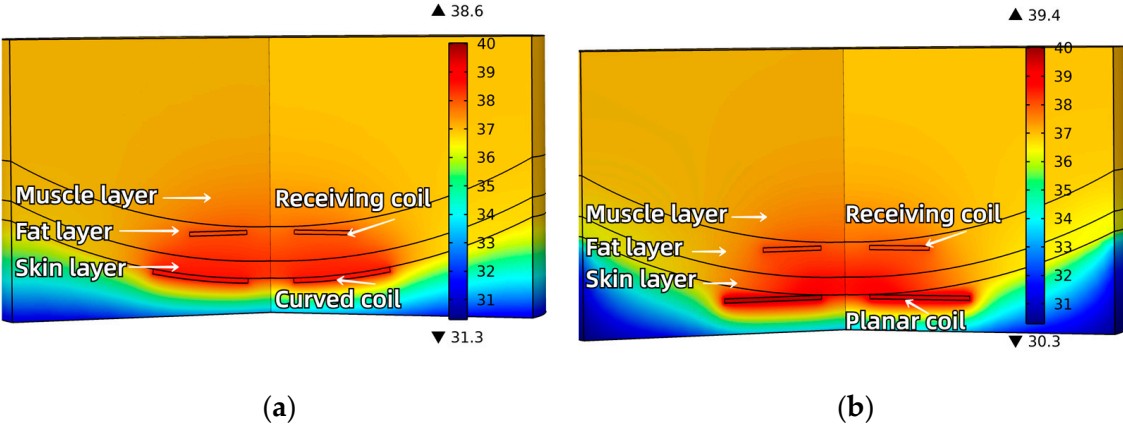

                       **(a)**                                      **(b)**

**Figure 10.** Temperature variation of coils: (**a**) $\alpha = 30°$ curved coil temperature; (**b**) $\alpha = 0°$ planar coil temperature.

In Figure 10, the temperature distribution of the transmitting and receiving coils after being powered for 30 min are illustrated. Specifically, as shown in Figure 10a, the temperature increase is more uniform in both the curved coil and the surrounding skin tissue. In contrast, Figure 10b shows that the planar coil exhibits more pronounced localized heating. Furthermore, the simulation results indicate that the peak temperatures for the curved and planar coils are 38.6 °C and 39.4 °C. Notably, the peak temperature of the curved coil differs by 2.1 °C from the initial temperature of the human tissue and blood, while the peak temperature of the planar coil differs by 2.9 °C. Therefore, for implantable medical devices that require prolonged use, the curved coils offer superior temperature performance compared to planar coils, potentially reducing the risk of thermal damage.

## 4. Experimental Verification

### 4.1. Experimental Platform

To validate the results of coil optimization performed in the finite element simulation software, an experimental platform was constructed, as shown in Figure 11. In the experiments, a microcontroller STM32 output complementary PWM signals to the full-bridge inverter circuit, and an electronic load VICTOR-3801S was used to simulate the load of an artificial heart. The main parameters of the system are presented in Table 1.

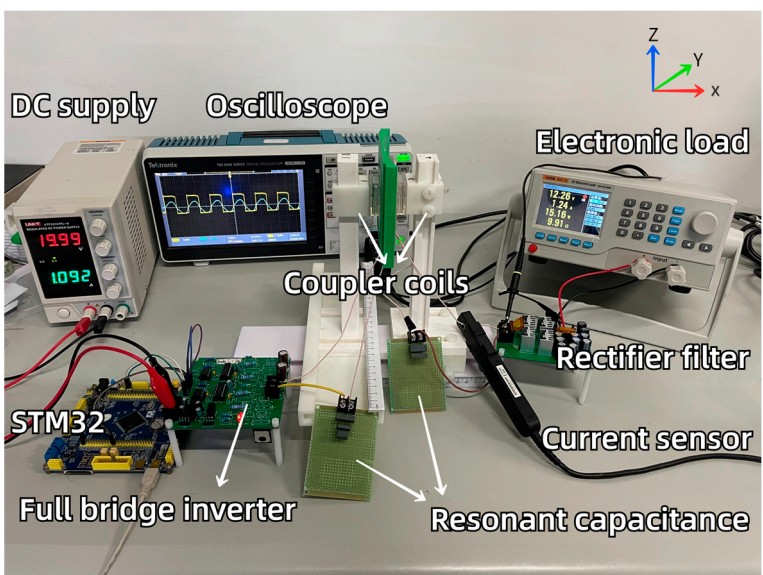

**Figure 11.** Experimental platform setup.

**Table 1.** Main parameters of the system.

| Parameter | Note | Value |
| --- | --- | --- |
| $U$ | Input voltage | 20 V |
| $f$ | Resonant frequency | 200 kHz |
| $L_R$ | Inductance of receiver coil | 11.52 uH |
| $C_2$ | Compensating capacitance | 54.97 nF |
| $R_L$ | Load | 10 Ω |

To validate the optimization simulation results of Section 2 from the perspective of publishing journal papers, coils with different parameters were designed based on the previous finite element simulations. Coils, as shown in Figure 12, were wound with varying turns, inner diameters, and degrees of bending. However, the specifications of the coils were all Litz wires, composed of 400 strands, each with a diameter of 0.04 mm. The coupling coils in the wireless power transfer system studied herein adopted an asymmetric design, meaning the transmitting and receiving coils were of different sizes. Among them, Coil

7 serves as the receiving coil, while Coils 1 to 6 serve as transmitting coils. The specific parameters of the coils are listed in Table 2.

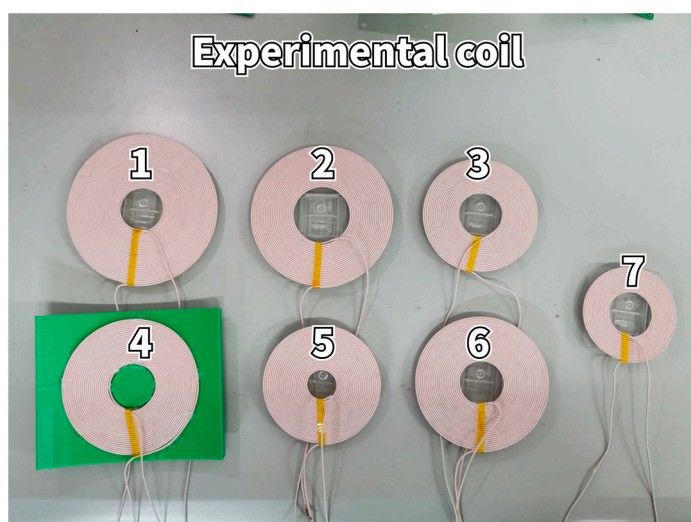

**Figure 12.** Experimental coils.

**Table 2.** Main parameters of the coils.

| Number | Turns | $D_{in}$ (mm) | $\alpha$ (°) | Inductance (μH) | Internal Resistance (mΩ) | $Q$ | Compensating Capacitance (nF) |
|--------|-------|---------------|--------------|-----------------|--------------------------|-----|-------------------------------|
| 1 | 30 | 30 | 0 | 58.12 | 392.5 | 186.3 | 10.81 |
| 2 | 25 | 40 | 0 | 44.95 | 318.3 | 171.0 | 14.09 |
| 3 | 20 | 30 | 0 | 24.19 | 198.3 | 153.9 | 26.27 |
| 4 | 25 | 30 | 30 | 35.54 | 261.3 | 170.4 | 17.81 |
| 5 | 25 | 20 | 0 | 27.11 | 212.5 | 160.8 | 22.73 |
| 6 | 25 | 30 | 0 | 35.43 | 252.5 | 173.6 | 17.82 |
| 7 | 15 | 30 | 0 | 11.52 | 112.1 | 128.3 | 54.97 |

*4.2. Result and Discussion*

To investigate the impact of coil turn count on the transmission characteristics of the system, Coils 1, 3, and 6 for transmission, along with Coil 7 for reception, were used in the experiment. By setting both *X*-axis and *Y*-axis misalignment distances ranging from 0 to 30 mm, with increments of 5 mm, the system's transmission efficiency was measured and calculated, with the efficiency with misalignment distance results displayed in Figure 13.

As illustrated in Figure 13a, when the receiving coil remains the same, the system efficiency for different numbers of turns initially increases and then decreases with the increasing misalignment distance along the *X*-axis. The increase in efficiency during the initial phase is due to the enhanced magnetic coupling resulting from the adjustment of coil positions, whereas the decrease in efficiency during the later phase is attributed to the excessive misalignment weakening the coupling. Additionally, the peak transmission efficiency for the three sets of coils occurs at different misalignment distances. Specifically, the 25-turn coil achieves the highest efficiency of 77% at an *X*-axis misalignment distance of 15 mm, while the 20-turn and 30-turn coils reach their respective peak efficiencies of only 73% and 74%.

To further analyze the performance of the coils under *Y*-axis misalignment, an experiment was conducted with an *X*-axis misalignment distance set at 15 mm, the condition for maximum efficiency. As shown in Figure 13b, the transmission efficiency of coils with different numbers of turns decreases with increasing *Y*-axis misalignment. The magnetic coupling gradually weakens with the increase in *Y*-axis misalignment, leading to a decrease

in transmission efficiency. Furthermore, the 25-turn coil consistently exhibits higher transmission efficiency compared to the 20-turn and 30-turn coils, indicating that the 25-turn coil has better anti-malposition capability under *Y*-axis misalignment.

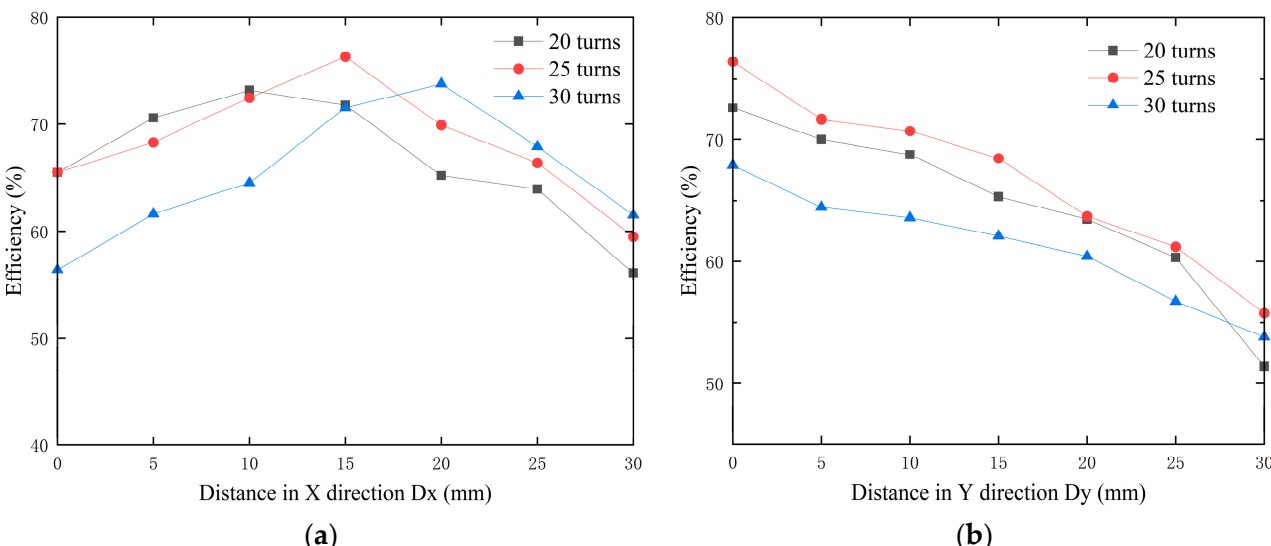

**Figure 13.** Efficiency with misalignment distances: (**a**) *X*-axis misalignment; (**b**) *Y*-axis misalignment.

In practical applications, the transmission distance of the system is around 15 mm. The 25-turn coil demonstrates high transmission efficiency and strong anti-malposition capability under various misalignment conditions. Therefore, the 25-turn coil was selected, and the experimental results were consistent with the simulation results.

To verify the influence of the inner diameter on the system's transmission characteristics, Coils 2, 5, and 6 were used as transmitting coils and Coil 7 as the receiving coil. The lateral misalignment distance was varied from 0 to 30 mm in 5 mm steps, and the system's transmission efficiency was calculated, with the results shown in Figure 14.

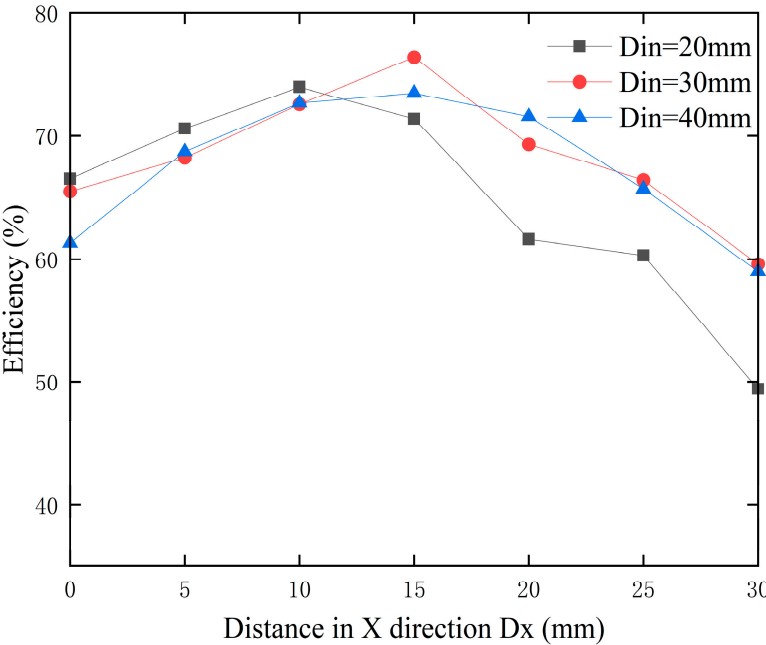

**Figure 14.** Relationship between transmission efficiency and *X*-axis misalignment distance.

As indicated by Figure 14, the transmission efficiency of transmitting coils with different inner diameters initially increases and then decreases with the increase in *X*-axis

misalignment distance. The coil with an inner diameter of 30 mm achieves the highest efficiency of 77% at a transmission distance of 15 mm, while the peak efficiencies for coils with inner diameters of 10 mm and 20 mm are only 72% and 73%, respectively. Therefore, using a transmitting coil with an inner diameter of 30 mm is preferable, and the experimental results are consistent with the simulation outcomes.

To assess the superior impact of curved coils on the system's transmission efficiency, Coils 4 and 6 were used as transmitting coils and Coil 7 as the receiving coil in the experiments. By varying both the lateral and longitudinal misalignment distances from 0 to 30 mm in 5 mm increments, the system's transmission efficiency was measured and calculated, with the results presented in Figure 15.

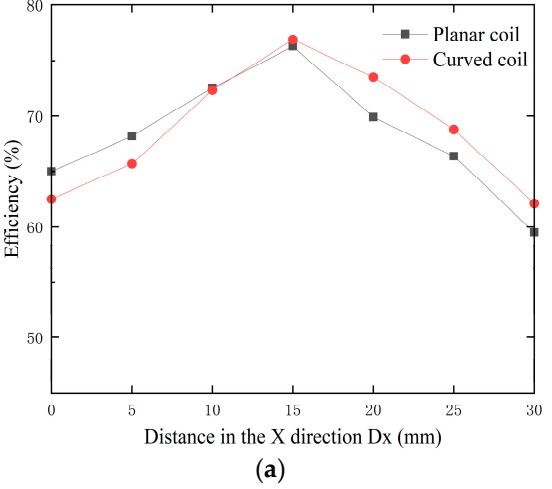

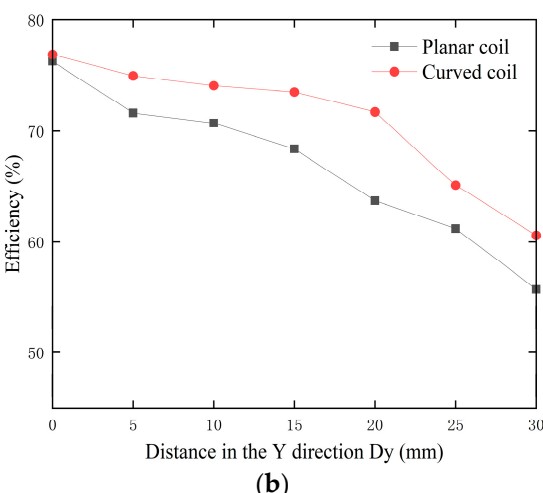

(**a**)  (**b**)

**Figure 15.** Effect of curved coil on system transmission efficiency: (**a**) *X*-axis misalignment; (**b**) *Y*-axis misalignment.

As shown in Figure 15, during *X*-axis misalignment, the transmission efficiency of the curved coils is slightly lower than that of the planar coils before 10 mm, but beyond 10 mm, the efficiency of the curved coils consistently exceeds that of the planar coils. When *Y*-axis misalignment occurs at $D_x = 15$ mm, the transmission efficiency of the curved coils is higher than that of the planar coils, and the efficiency changes more steadily within the $D_y = 0$–20 mm range. Therefore, curved coils demonstrate superior resistance to misalignment compared to planar coils. Figure 16 shows the voltage and current signals at the transmitting and receiving ends for the curved coil at $D_x = 15$ mm, illustrating that both voltage and current signals are in phase when the system resonates.

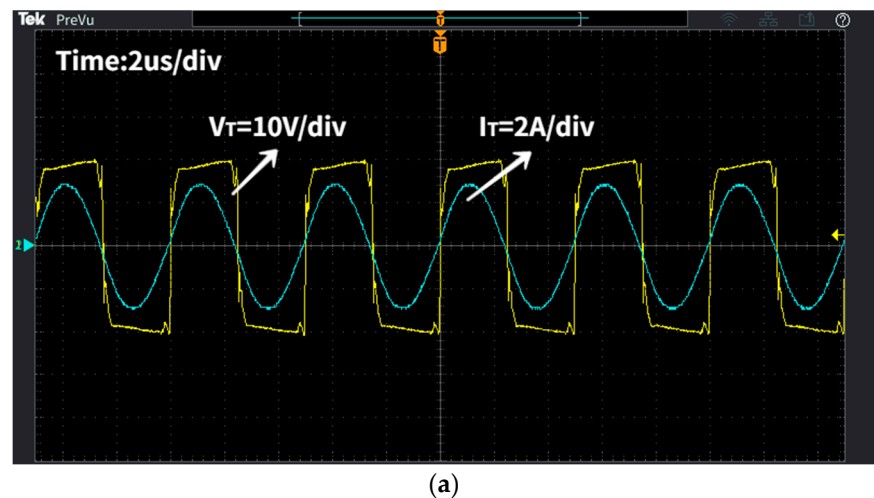

(**a**)

**Figure 16.** *Cont.*

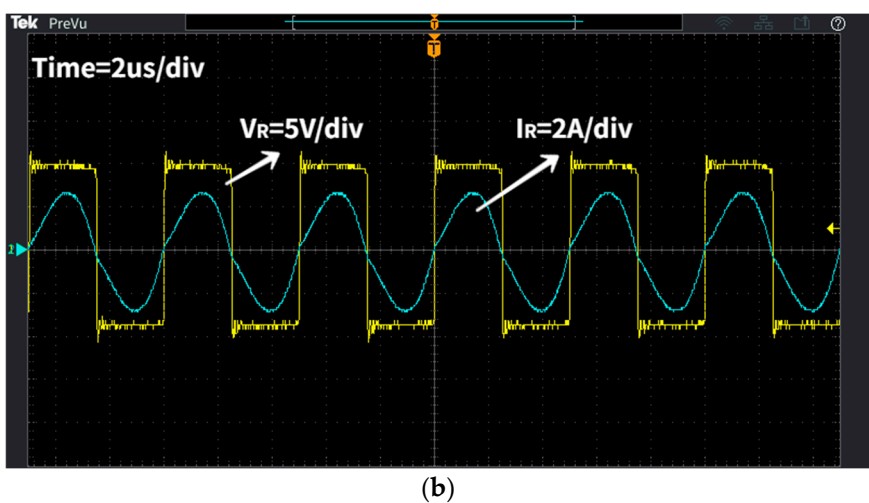

(**b**)

**Figure 16.** Voltage and current signals of the coils: (**a**) transmitting side signal; (**b**) receiving side signal.

To verify the temperature rise of curved and planar coils, the experiment utilized pork ribs, which are similar to the human thoracic cavity, as a biological medium with a central thickness of approximately 15 mm. The coil temperature measurement experiment is shown in Figure 17, with the receiving coil placed under the biological tissue. The temperature measurement experiment used a YET-610L single-channel thermocouple thermometer, which offers a temperature measurement range from $-200$ °C to 1370 °C, a measurement resolution of 0.01 °C, and an accuracy error of $\pm0.4$ °C. The K-type thin-film thermocouple probe used can closely conform to the surface of the biological tissue, allowing for more precise temperature measurements.

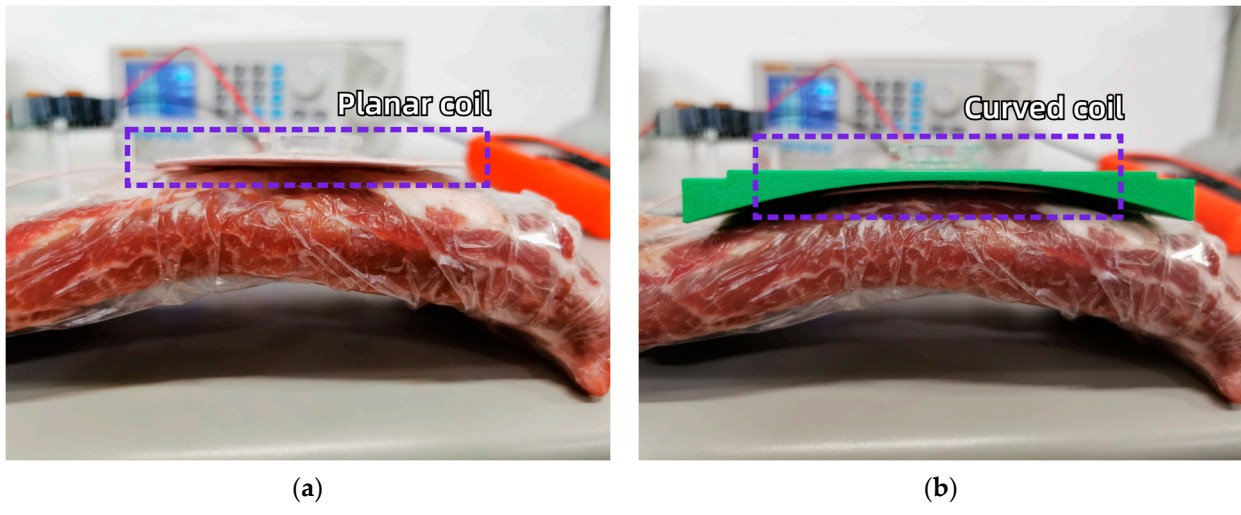

(**a**)                          (**b**)

**Figure 17.** Coil temperature measurement experiment: (**a**) planar coil; (**b**) curved coil.

The experimental circuit parameters are as shown in Table 1. Coils 4 and 6 were used as transmitting coils, with an initial room temperature of 25 °C. Power was continuously supplied for 40 min, and the temperature of the coils was recorded every 5 min. The temperature rise of the two coils is illustrated in Figure 18.

As clearly visible from Figure 18, the temperature of both coils gradually increases over time and the temperature curve stabilizes after about 30 min of continuous power supply. The planar coil reached its peak temperature of 29.21 °C after 40 min of electrification, while the peak temperature for the curved coil was 28.7 °C. The temperature curve of the curved coil was slightly lower than that of the planar coil, indicating that replacing traditional planar coils with curved coils can reduce the system's heat generation. Notably, compared

to the coil temperature simulations in Section 2, the experimental temperature rise was slightly higher. In the simulations, the highest temperature increase for the curved coil was 2.1 °C, whereas in the actual experiments it reached 3.7 °C. The primary reason for this discrepancy is that the COMSOL Multiphysics coupling considered blood perfusion rates and the specific heat capacity of blood; hence, the simulated temperatures were lower than those using pork tissue in experiments.

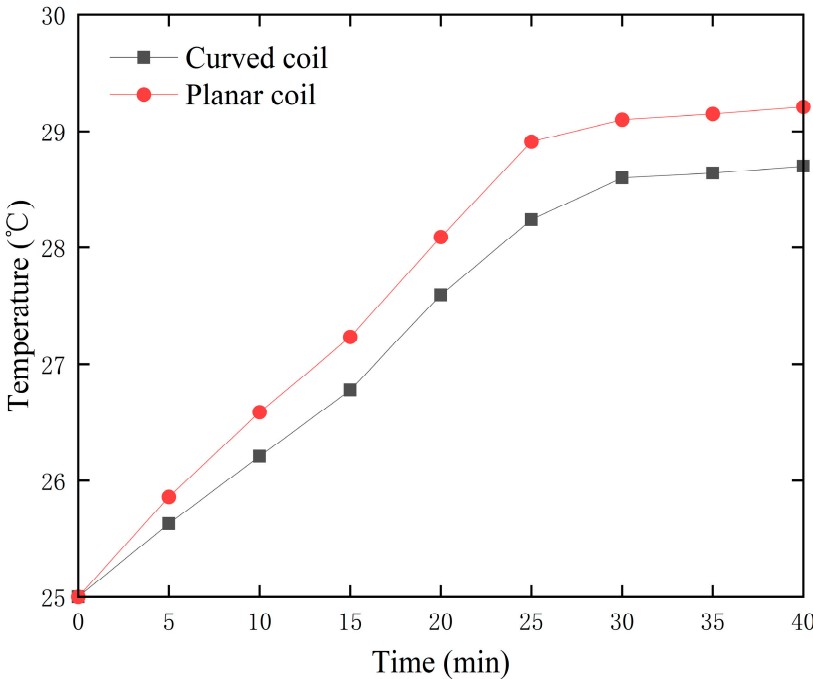

**Figure 18.** Temperature variation of different types of coils.

## 5. Conclusions

This study is dedicated to addressing the challenge of ensuring a stable energy supply for artificial hearts by optimizing the design of coils within TETS, thereby facilitating efficient and safe power delivery to implants. A comprehensive analysis was undertaken of several crucial parameters of the transmitting coil, including inner diameter, number of turns, and turn pitch, as well as the degree of coil curvature. Through simulations conducted in COMSOL, these parameters were optimized to maximize both the coupling coefficient and the system's transmission efficiency. Additionally, the issue of temperature increases during practical application of the coils was addressed, assessing their potential safety implications for biological tissues. Rigorous experimental validations were performed using the coils that were constructed, leading to significant findings:

The impact of variations in the inner diameter, number of turns, and turn pitch on the coupling coefficient and transmission efficiency of the TETS was investigated. It was found that coils with a 30 mm inner diameter and a turn pitch of 0 mm, when wound tightly and maintaining the same number of turns, achieved the best balance between coupling efficiency and transmission efficiency, while demonstrating a strong magnetic field distribution. Furthermore, a comparative analysis between curved and planar coils was conducted. The results from both simulations and experiments showed that, compared to planar coils, curved coils exhibited a higher coupling efficiency and greater resistance to misalignment, making them more suitable for alignment with the human thoracic cavity in practical uses. Moreover, during a 40 min electrification period, curved coils were observed to maintain better temperature control, which is crucial for medical implants that require long-term stability and effectively reduces the risk of thermal damage to human tissues.

These coil optimization strategies have not only enhanced the transmission efficiency and safety of the TETS but also provided substantial theoretical and practical insights for

the design of similar medical implant devices. Future work will continue to explore other potential optimization parameters to further refine the system design, ensuring higher energy utilization efficiency and improved biocompatibility.

**Author Contributions:** Conceptualization, R.W.; methodology, L.W. and F.H.; software, H.L. and J.C.; validation, H.L. and Q.L.; formal analysis, F.H.; writing – original draft, Q.L.; writing – review & editing, Y.F.; visualization, J.C.; supervision, Y.F.; funding acquisition, R.W. All authors have read and agreed to the published version of the manuscript.

**Funding:** This research was funded by Natural Science Foundation of Zhejiang Province (grant number LGG20F020008) and this work was supported by the "Pioneer" and "Leading Goose" R&D Program of Zhejiang (grant number 2023C01056).

**Data Availability Statement:** The data presented in this study are available on request from the corresponding author.

**Conflicts of Interest:** The authors declare no conflict of interest.

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
