# Peer review of "Design and Optimization of Coil for Transcutaneous Energy Transmission System"

_electronics, doi:10.3390/electronics13112157_

Round 1
Reviewer 1 Report
Comments and Suggestions for Authors
The present article deals with a topical issue from an interdisciplinary field. The authors present the Design and Optimization of the Coil for a Transcutaneous Energy Transmission System. This beautiful area is currently receiving much attention. The authors focus their paper on a view of the situation from the point of view of circuit theory, present a circuit model, and its analytical solution. The main idea and the core of the paper can be described as the investigation/optimization of the realization of the whole excitation/acceptance system, both in terms of the values of the individual passive elements of the system and also in terms of the spatial arrangement of the individual inductors, which are coupled with the inductive coupling. The authors present the results obtained at the level of numerical modelling /COMSOL/ and compare them with experimental data. At the same time, they also address the issue of Joule losses, which is paramount in the biomedical field and must be given great attention.
The structure of the paper is clear and logical, which is a strong foundation for further improvement. I believe that with the necessary revisions, the paper has the potential to become an even more valuable contribution to the field.
I have reservations about the presentation of the results and the terminology/ shortcomings in the launch of the circuit model; see my comments below. The authors did much work, and their results are convergent. However, the conclusions need to be corrected, respectively. To give them correctly from a formal point of view.
Conclusion: To improve the quality of the article from a professional point of view and thus make it more attractive to the reader, I recommend that the authors adjust the article in accordance with Major Revision.
Fig.2: In the wiring diagram, it is necessary to repair:
1. Remove symbols + A - for power supply.
2. To LT and LR inductors supplement the labelling of the beginnings/ends of the windings
3. Remove symbols + and - at RL load
4. To the perimeter variables, it is necessary to add a symbol (dot above quantity) because these are phases
Fig. 4 and more: The pictures do not show how the quantity is displayed or even a range. This is understandable for the stakeholder without explanation, but in general, it may not be so. Please add or indicate the scale outside the image itself.
Figure 6: Individual adjacent functional values cannot be connected to a line. Word on the horizontal axis "Turns" please change to "Number of Turns, n [-]"Figure 7 and others: adjacent functional values cannot be connected to a straight line.
Figure 9: The picture is confusing. Labels to specify the situation are necessary. Overall, the article requires a revision of professional terms and specific quantities according to electrical practice.
Reviewer 2 Report
Comments and Suggestions for Authors
I would like to extend my gratitude to the authors for their diligent work provided on this manuscript. As I proceed to share my detailed review and feedback comments, I am mindful of the rigorous effort invested in your study and the potential impact of your findings.
Summary:
The authors present a wireless energy transmission system based on induction coupling trying to optimize the transmitter coil in terms of its inner diameter, number of turns, pitch, curvilinear geometries and temperature increase to find the optimal design parameters in simulation followed by benchtop testing. Though authors have tried to cover various parameters, I believe the manuscript needs improvement in its figures, data analysis and discussions instead of visually reporting that data in each graph. Lastly, the authors should discuss the impact of results of each parameter optimized on final application i.e. wirelessly powering artificial hearts eg: As an example, how does saving 0.5C of temperature increase impact or influence performance of the artificial heart implant or its environment and why that is important.
Comments:
1. Figure 4 and its related text is not clear in terms of its reasoning and conclusion. The scales on each simulation is not visible to naked eye (should be turned into white text). Also, authors suggest coil exhibits highest maximum field strength at Din = 10mm when its not clear what is the reasoning there. Furthermore, if the decision to finalize inner coil diameter is based on Figure 5, then what value does Figure 4 add to the argument?
2. In Figure 8, pitch should be linked to Dgap in Figure3.
3. Figure 10 has similar issues as with Figure 4 (Comment 1 above)
4. Figure 11, it must be clearly highlighted what is x, y and z direction. Furthermore, in related text the improvement of curved coils must be quantified, as each curve seems to have more or less equal sensitivity to lateral movement. The marginal improvement, if any, must be quantified and discussed.
5. Figure 12, the simulations setup is not clear as to what was normalized between the two simulations? Was it energy transferred? Furthermore, there is no explanation or analysis for the behavior seen.
6. Figure 15, the analysis seems wrong. Authors should clearly define what they mean by lateral vs longitudinal misalignment. In Line 331, does "transmission distance of 15mm" mean same as "misalignment distance" represented in Figure 15, as stated in Line 323. Furthermore, authors claim that 25 turn coil consistently outperformed the 20 turn coil under lateral misalignment, when its evident from the graph that is not the case at misalignment distances of 20mm or higher. Its also not clear why authors are not including 30 turns data in Figure 15b?
7. Figure 20, the marginal improvement (<0.5C) in temperature seems unlikely to make much difference considering that according to NIH, body temperatures can vary between 0.25C to 0.5C every day.
Comments on the Quality of English LanguageCertain writings in the text sounds confusing or contradictory, which may be due to language errors.
Reviewer 3 Report
Comments and Suggestions for Authors
The manuscript describes coil design for inductive power transmission through a biological material. Authors present theory, simulation, and experiments in an easy-to-understand writing style. Experiments show maximum power transmission efficiency exceeding 75% through 15 mm inter-coil distance.
The following are this reviewer's comments.
1. Coil design:
Coil design has been described in details. Please add the following information.
1) Number of the Litz wire strands.
2) Coil quality factor or R1/R2 in Figure 2.
3) Transmission efficiency due to Tx and Rx coils only. As you know, there is a formula for this task.
4) Coils are designed for 200-kHz operation. It is not known that the designed coil's efficiency is maximum at 200 kHz. Please add the coil efficiency versus frequency to find out the optimum WPT frequency.
5) Ferrite backing is usually employed in inductive WPT for increased coupling coefficient and for shielding the magnetic field. It appears that a ferrite backing is not used. Please comment on this.
2. Figure 15 and 17:
Figure 3 shows the x and y axes. Due to geometrical symmetry, misalignment in x and y directions should be the same but Figures 15 and 17 tell a different story. Please check.
3. Coil efficient measurement requited:
Figures 15-17 show the WPT system efficiency exceeding 75%. The achieved efficiency consists of the circuit efficiency and the coil efficiency. The manuscript is about the coil design. So the coil efficiency measurement is important. Coil efficiency can be measured by transmitting a 200-kHz sinusoidal signal with 10-ohm load and compensating capacitors. If a ferrite backing has been used, the coil current needs to be same as in the actual WPT system.
4. Figure 19:
1) Please specify the thickness of port ribs. Is it the same 15 mm?
2) The article is about WPT through a biological material. The power loss or efficiency reduction due to the biological material is important data. Please check the efficiency reduction due to port ribs by comparing the system efficiency with pork ribs and without pork ribs (air or low loss material like Styrofoam).
Round 2
Reviewer 1 Report
Comments and Suggestions for Authors
The authors have submitted a revised version of the original article for consideration and comments on accepting comments. After reviewing the article, I conclude that the article is qualitatively on a higher level. The authors have incorporated the necessary changes to publish the article in a high-quality journal. There are still some minor formal inaccuracies in the article that lower its level but do not diminish its quality. Based on a review of this revised version, I recommend the article for publication.